# Quantitative Determination of Acetamiprid in Pollen Based on a Sensitive Enzyme-Linked Immunosorbent Assay

**DOI:** 10.3390/molecules24071265

**Published:** 2019-04-01

**Authors:** Qingkui Fang, Quan Zu, Xiude Hua, Pei Lv, Wanwen Lin, Dahe Zhou, Zihan Xu, Jiarui Fan, Xiaohan Li, Haiqun Cao

**Affiliations:** 1School of Plant Protection, Anhui Agricultural University, Hefei 230036, China; qkfang@163.com (Q.F.); zuquan06053122@163.com (Q.Z.); mei.mei906888037@163.com (W.L.); 17352900322@163.com (D.Z.); zhmirror03@sina.com (Z.X.); 15156879285@163.com (J.F.); lixianhan99@sina.com (X.L.); 2Key Laboratory of Biology and Sustainable Management of Plant Diseases and Pests of Anhui Higher Education Institutes, Anhui Agricultural University, Hefei 230036, China; 3College of Plant Protection, Nanjing Agricultural University, Nanjing 210095, China; huaxiude@njau.edu.cn; 4School of Resource & Environment, Anhui Agricultural University, Hefei 230036, China; lvpei@ahau.edu.cn

**Keywords:** pollen, acetamiprid, pesticide, heterogeneous coating antigen, biotinylated mAb, Bic-ELISA

## Abstract

A sensitive biotinylated indirect competitive enzyme-linked immunosorbent assay (Bic-ELISA) was developed to detect acetamiprid pesticides in pollen, based on the heterogeneous coating antigen and biotinylated anti-acetamiprid monoclonal antibody. Under optimized experimental conditions, the detection limit for the Bic-ELISA was 0.17 ng/mL and the linear range was 0.25–25 ng/mL. The cross-reactivities could be regarded as negligible for the biotinylated antibodies with their analogues except for thiacloprid (1.66%). Analyte recoveries for extracts of spiked pollen (camellia pollen, lotus pollen, rape pollen) ranged from 81.1% to 108.0%, with intra-day relative standard deviations (RSDs) of 4.8% to 10.9%, and the average reproducibility was 85.4% to 110.9% with inter-assay and inter-assay RSDs of 6.1% to 11.7%. The results of Bic-ELISA methods for the Taobao’s website samples were largely consistent with HPLC-MS/MS. Therefore, the established Bic-ELISA methods would be conducive to the monitoring of acetamiprid in pollen.

## 1. Introduction

Pollen is a fine to coarse powdery substance, and comprises pollen grains, which are the male microgametophytes of seed plants, producing male gametes (sperm cells) [1]. It is a good, nutritious food and tonic, and has been favored by consumers, especially female consumers, because it is rich in essential nutritional components such as carotenoids, proteins, flavonoids, plant sterols, phytosterols, and active proteases [2,3,4]. Pharmacological studies have shown that active proteases in pollen can decrease blood lipids, enhancing the immune function of the human body to prevent aging; additionally, it has a certain effect on freckle beauty [5]. However, pollen plants suffer from pests and diseases during the planting process [6,7]. Farmers usually use chemicals to control these, thus polluting the pollen and forming a threat to human health and environmental safety.

Over the past decades, the neonicotinoids have been the most widely used insecticide globally [8], acting on soil as well as branches and leaves [9,10]. This systemic insecticide has a unique mode of action of a nicotinic acetylcholine receptor (nAChR) agonist [11], and it is very effective in controlling pests such as aphids, whitefly, thrips, beetles, leafhoppers, and borers [12]. Therefore, acetamiprid is widely used in the control of pollen plant pests. However, the neonicotinoids can have an impact on human health, for example, human exposure to 2,4-D can occur via soil ingestion, inhalation, and dermal contact, and result in some adverse health effects [13,14]. Currently, there are several methods for detecting residues of acetamiprid in pollen, such as GC [15], GC-MS [16], HPLC [17], UPLC/q-TOF MS [18], and HPLC-MS/MS [19,20,21]. These residual analysis methods have superior sensitivity and reliability. However, in experiments, the pretreatment of samples takes a long time and incurs significant costs, requires professional personnel to perform, involves expensive instrumentation, and is not conducive to market and field testing [22,23,24]. Therefore, we urgently need to develop a rapid, easy and high-throughput analysis technology for field detection.

Immunoassay is a user-friendly analytical method that is relatively fast, simple, and economic [25]. There have been several immunoassay methods developed to detect acetamiprid, such as a commercial enzyme-linked immunosorbent assay (ELISA) to determine acetamiprid for residue analysis in peach, apple, strawberry, cucumber, eggplant, and tomato [26]. A direct competitive ELISA is used for the quantitative detection of acetamiprid residues in spinach, welsh onion and Chinese chive [27], and a competitive ELISAs were developed to detect acetamiprid and imidacloprid in cucumber, green pepper, tomato and apple [28]. However, to the best of our knowledge, there is a lack of studies on the immunoassay of chloronicotinyl neonicotinoid insecticide residues in pollen [29,30,31]. This will not be conducive to the market supervision of pollen samples.

Previous work showed that acetamiprid has the highest detection rate of chloronicotinyl neonicotinoid insecticides in pollen samples, reaching 1.7%, and the detected concentration ranged from 5.2 ng/g from 63.6 ng/g [32]. Because pollen is an important bee food and human nutriment, we need a fast, sensitive and high-throughput detection method to ensure the safety of bees and humans. Therefore, the aim of this research was to develop a biotinylated indirect competitive ELISA (Bic-ELISA) for detecting acetamiprid pesticides in pollen. Through the preparation of coating antigens, biotinylation of anti-acetamiprid monoclonal antibody, optimized experimental conditions, and the removal of the pollen matrix, we successfully established a Bic-ELISA method for the detection of acetamiprid residues in pollen.

## 2. Results and Discussion

### 2.1. Verification of Hapten

Hapten H1 and H2 were synthesized, and the structure was clarified by ^1^H-NMR and ESI-MS. The results of ^1^H-NMR and ESI-MS were described as follows:

Hapten H1: ^1^H-NMR (600 MHz, DMSO-d6) δ 2.38 (s, 3H), 2.59 (t, *J* = 12 Hz, 2H), 3.08 (s, 3H), 3.27 (t, *J* = 12 Hz, 2H), 4.63 (s, 3H), 7.26 (d, *J* = 8.3 Hz, 1H), 7.51–54 (m, 1H), 8.35 (s, 1H), δ 10.11 (s, 1H). ESI-MS: 292.02.

Hapten H2: ^1^H-NMR (600 MHz, CDCl_3_) δ 2.81 (t, *J* = 6.8 Hz, 2H), 2.87 (s, 3H), 3.39 (t, *J* = 6.8 Hz, 2H), 4.57 (s, 2H), 4.72 (s, 2H), 4.76 (s, 2H), 7.47 (s, 1H), 8.42 (s, 1H). ESI-MS: 362.05.

The results showed that acetamiprid hapten was successfully synthesized by one step method. The carboxylic acid moieties of hapten will facilitate the binding with carrier proteins to synthesize artificial antigens.

### 2.2. Optimization of the Bic-ELCIA

There are many parameters that may affect the binding of the antibody to the analytes. In our study, H1-OVA and H2-OVA, the coating antigen and BAb concentration, ionic strength (0–1.6 mol·L^−1^), and pH (6.5–9.0) of the Bic-ELISA system were optimized. The coating antigen of H1-OVA and H2-OVA were investigated using the noncompetitive and the competitive ELISA, the noncompetitive ELISA indicated that H1-OVA and H2-OVA had a higher titer, and the competitive ELISA was evaluated to select the best sensitivity through the half-maximal inhibition concentration values [IC50 values (ng/mL) of the combination between BAb and coating antigen]. The result shows that the IC50 value of Bab/H1-OVA and Bab/H2-OVA were more than 1 μg/mL and 3.2 ng/mL. Therefore, the H2-OVA were selected for subsequent Bic-ELISA research. Second, in this study, the best working concentration of the coating antigen and Bab were 2.6 μg/mL and 1.2 μg/mL separately, determined by using checkerboard titration. Figure 1 shows the results of ionic strength (Figure 1A) and pH (Figure 1B) of the Bic-ELISA system. The main criterion for evaluating the Bic-ELISA assay was the highest ratio of OD450_max_/IC50 [33]. Based on the results of Figure 1, the optimized ionic strength and pH conditions for the Bic-ELISA were selected as the ionic strength was 0.4 mol/L, and pH = 8.0.

### 2.3. Analytical Bic-ELISA for Acetamiprid

Under optimized experimental conditions, the Bic-ELISA analytical performance for the acetamiprid detection was examined with different concentrations (0.025, 0.05, 0.1, 0.25, 0.5, 1.0 2.5, 5.0, 10, 25, 50, 100 ng/mL) of standard acetamiprid in PBST. The results presented in Figure 2 indicated that the developed Bic-ELISA was suitable for the determination of acetamiprid. In Figure 2A, the graph between the acetamiprid concentration and binding (B/B_0,_ B and B_0_ are the absorbances of the analyte presence and absence, respectively) was plotted. After the conversion of Figure 2A, it was observed that, in Figure 2B, the graph between the logarithm of acetamiprid (ng/mL) concentration and B/B0 was linear in the range of 0.25–25 ng/mL, and the regression equation was y = −0.4102x + 0.5978, R^2^ = 0.9908. The limit of detection (LOD) was 0.17 ng/mL by the extrapolation of B0-2SD.

### 2.4. Cross-Reactivity

Acetamiprid, thiacloprid, thiamethoxam, imidacloprid, dinotefuran, nitenpyram, clothianidin and 2,4-D were tested for detecting the specificity of the optimized Bic-ELISA. The results of cross-reactivity were shown in Table 1, the highest cross-reactivity was 1.66%, obtained from thiacloprid. Meanwhile, negligible cross-reactivity (<0.5%) with thiamethoxam, imidacloprid, dinotefuran, nitenpyram, clothianidin and 2,4-D was observed. These results demonstrated that the developed Bic-ELISA had a high specificity and might be applied for the acetamiprid determination.

### 2.5. Accuracy

The matrix interference of camellia pollen, lotus pollen, and rape pollen on the established Bic-ELISA was investigated. To evaluate the influence of the pollen matrix on the Bic-ELISA, camellia pollen, lotus pollen, and rape pollen were extracted, and diluted at 4-fold for three pollen samples.

The average recoveries and relative standard deviations (RSDs) of acetamiprid in three pollen matrices at 10, 20 and 50 ng/g, and on three consecutive days are shown in Table 2. The average recoveries were 81.1–108.0% with intra-assay RSDs (*n* = 3) of 4.8%–10.9% and inter-assay RSDs (*n* = 3) of 6.111.7%. Thus, the recoveries and reproducibility of the Bic-ELISA in our cases were acceptable.

### 2.6. Analysis of Authentic Samples

Camellia pollen, lotus pollen, and rape pollen were purchased based on the ranking of sales from high to low on Taobao’s website, and we chose the top five pollens in each ranking. According to the results of detection, neither the Bic-ELISA method nor HPLC-MS/MS method detected acetamiprid residues. Based on our research results, the linear detection range of acetamiprid was 5–200 ng/g in the HPLC-MS/MS method [32], and the linear detection range of acetamiprid was 2–200 ng/g in the Bic-ELISA method. That is to say, the residues of acetamiprid in 15 pollen samples were less than the detection range in both methods. However, in crude pollen, the acetamiprid detection rate was 1.7%, and the detected concentration ranged from 5.2 ng/g to 63.6 ng/g [32]. There are two possible reasons for this: on the one hand, there may not be acetamiprid residue in the pollen samples we purchased; on the other hand, pesticide residues in agricultural products during processing will be subjected to a certain degree of digestion [34,35,36], due to the crude pollen need for drying and sterilization, so that the residual acetamiprid degradation is beyond the linear range. Thus, we might try to study effect of processing factors on acetamiprid residues in pollen in future studies.

## 3. Materials and Methods

### 3.1. Reagents

The anti-imidacloprid monoclonal antibody was produced in the State & Local Joint Engineering Research Center of Green Pesticide Invention and Application, Nanjing 210095, China [37]. Acetamiprid, thiacloprid, thiamethoxam, imidacloprid, dinotefuran, nitenpyram, clothianidin and 2,4-D were purchased from Dr. Ehrenstorfer (Augsburg, Germany). Horseradish peroxidase (HRP) Conjugated Streptavidin was purchased from Boster Biological Technology Co., Ltd. (Wuhan, China). (+)-biotin *N*-hydroxysuccinimide ester (BNHS), polyoxyethylene sorbitan monolaurate (Tween-20), 3,3′,5,5′-tetramethylbenzidine (TMB), and Ovalbumin (OVA) were purchased from Sigma Chemical Co. (St. Louis, MO, USA). Chemical reagents such as 3-mercaptopropionic acid (3-MPA), dimethyl sulfoxide (DMSO), H_2_O_2_, and others were all purchased from Aladdin (Shanghai, China).

### 3.2. Instruments and Equipment

Polystyrene 96-well microtiter plates were purchased from Costar (Corning, Tewksbury, MA, USA), and washed with ELX405^TM^ (BioTek, Winooski, VT, USA). The coating antigen and antibody protein concentration was determined by a Nanodrop 1000 UV-VIA (Thermo, Waltham, MA, USA). The optical density value of 450 nm ultraviolet wavelength (OD450) was measured with MULTISKAN GO (Thermo Scientific, Boston, MA, USA). Samples were vortex mixed with Genius 3 (IKA, Germany). Centrifugation was performed with a JW-1012 low-speed centrifuge (Jiawen, Hefei, China). HPLC-MS/MS was performed using a Waters ACQUITY UPLC Xevo TQMS instrument (Waters Corp, Milford, MA, USA).

### 3.3. Preparation of Coating Antigens

The acetamiprid hapten synthetic route [38] is shown in Figure 3. A mixture of 0.45 g of KOH, 0.42 g of 3-MPA and 1.02 g of acetamiprid (or thiamethoxam) were dissolved in 20 mL of ethanol, and then stirred at 80 °C for 2 h. Following the mixture was filtered and concentrated. The residue was dissolved in 50 mL of pure water and adjusted to pH 2 using 1 mol/L HCl. The solution was extracted 3 times with 30 mL of ethyl acetate. The extract was washed 3 times with 30 mL of water, dried over anhydrous Na_2_SO_4_, and concentrated. The product was recrystallized using methanol to yield a white solid, which was characterized by NMR.

The coating antigens were prepared using the active ester method [39] as described previously, H1 (synthesized using acetamiprid) and H2 (synthesized using thiamethoxam) were covalently attached through their carboxylic acid moieties to the lysine groups of OVA. The coating antigens of H1–OVA and H2–OVA were purified by dialysis in phosphate buffer saline (0.15 mol/L, pH 7.4).

### 3.4. Biotinylation of Anti-Acetamiprid mAb

Biotinylated anti-acetamiprid mAb was prepared as described previously [40] with some modifications. Briefly, first, anti-acetamiprid mAb was dialyzed against Carbonate-buffered saline (0.05 mol/L, pH 9.6) for 4 h. Second, BNHS was dissolved in DMSO, and then the dialysis solution was added a 20-fold molar excess of anti-acetamiprid mAb (5.0 mg/mL). And finally, the solution was thoroughly stirred at RT for 6 h. Following the biotinylated anti-acetamiprid mAb (BAb) was dialyzed against 0.01 mol/L phosphate buffer saline, and then stored at 4 °C until required.

### 3.5. Performance of Biotinylated Indirect Competitive Enzyme-Linked immunosorbent Assay (Bic-ELISA)

The basic steps in the Bic-ELISA process are as follows. Microtiter plates with 96 wells were coated with optimized concentrations of H–OVA in Carbonate-buffered saline (0.05 mol/L, pH 9.6) (50 μL each well) by incubation for 2 h. Plates were then blocked with 1% gelatin in phosphate-buffered saline (0.15 mol/L, pH 7.4) (100 μL each well) by incubation for 1.5 h. Aliquots of 25 μL for each well of analyte were dissolved in working solution and 25 μL for each well of biotinylated anti-acetamiprid mAb (BAb) (diluted with working solution) at a previously optimized concentration. After incubating for 1 h, 50 μL of each well of diluted (1/10,000) HRP–Conjugated Streptavidin in phosphate-buffered saline containing 0.05% Tween-20 (PBST) was added to the plate. The mixture was incubated for 1 h, followed by the addition of 50 μL to each well of a TMB solution (contained 0.4 mmol/L TMB and 3 mmol/L H_2_O_2_ in citrate buffer (pH 5.0)). After incubating for 15 min, the reaction was stopped by adding 25 μL 2 mol/L H_2_SO_4_, and the TMB termination solution absorbance was measured at 450 nm. All the 96-well microtiter plates were washed five times with PBST (0.15 mol/L phosphate buffer saline containing 0.05% Tween 20, pH 7.4) after each incubation, and incubations were performed at 37 °C, unless specified otherwise.

### 3.6. Immunoassay Optimization

The selection of the coating antigen/biotinylated anti-acetamiprid mAb (BAb) combination was performed by Bic-ELISA. The optimum combination of coating antigen (H1-OVA, H2-OVA) and BAb was confirmed based on the sensitivity of the Bic-ELISA, and the best working concentration of antigen and BAb determined by checkerboard titration. The dilution ratios of the Bab ranged from 1:200 to 1:25,600 and coating antigen from 1:500 to 1:8000 were confirmed based on the sensitivity of the Bic-ELISA.

The working solutions, used to dilute the acetamiprid standards and were tested using the established Bic-ELISA, were prepared at a series of pH values (ranging from pH 6.5 to pH 9.0) and ionic strengths (from 0.1 to 1.6 mol/L). Selection of Bic-ELISA test working solution with the best sensitivity as the optimization of working solution.

### 3.7. Cross-Reactivity

Cross-reactivity (CR) was studied for evaluating the selectivity of the Bic-ELISA, using the standard solution of the acetamiprid and thiacloprid, thiamethoxam, imidacloprid, dinotefuran, nitenpyram, clothianidin, and 2,4-D. The CR values were calculated as follows: CR % = (IC_50_ of acetamiprid/IC_50_ of analogue) × 100. Here, the CR of acetamiprid was defined as 100%.

### 3.8. Recovery

The performance of the Bic-ELISA was evaluated by the recovery of spiked pollen samples. Three different pollen samples (camellia pollen, lotus pollen, rape pollen) were chosen to evaluate the accuracy and precision of the Bic-ELISA. Before undertaking spiking and recovery studies, all pollen samples were verified without acetamiprid by HPLC-MS/MS.

One gram samples were placed in a centrifuge tube I (15 mL) and spiked with known concentrations of acetamiprid standard solution. The samples were thoroughly mixed by vortex and allowed to stand at RT for 1 h. Two and a half milliliters of pure water and 2.5 mL of acetonitrile was added to sample and the samples were shaken thoroughly for 3 min, and then 0.85 g filler A (anhydrous MgSO_4_: NaOAc = 4: 1) was added, thoroughly shaken for 30 s, and centrifuged at 4500× *g* for 5 min. One milliliter of the supernatants was then transferred to a centrifuge tube II (15 mL), containing 0.75 g filler B (anhydrous MgSO_4_: graphitized carbon black: primary secondary amine: octadecyl-bonded silica = 3: 0.15: 1: 1). Following this, 2.5 mL of acetonitrile was added to the centrifuge tube I, thoroughly shaken for 3 min, and centrifuged at 4500× *g* for 5 min. Two milliliters of the supernatant was transferred to centrifuge tube II and thoroughly shaken for 3 min, and centrifuged at 4500× *g* for 5 min. Finally, 2 mL of the extract was sampled in a glass cone (10 mL) and was evaporated at 30 °C until dried under a gentle stream of nitrogen. One milliliter of optimal working buffer (10% methanol) was added for Bic-ELISA analysis.

### 3.9. Real Sample

Authentic samples (including camellia pollen, lotus pollen, and rape pollen) were collected from Taobao-Online retailers. The Bic-ELISA was utilized to investigate the acetamiprid residues in purchased pollen samples. In addition, all the pollen samples were further confirmed by HPLC-MS/MS. The process of HPLC-MS/MS was performed as described previously [32].

## 4. Conclusions

In summary, this work presents the sensitive Bic-ELISA method for detecting acetamiprid pesticides in pollen samples based on a heterogeneous coating antigen and biotinylated anti-acetamiprid mAb (BAb). Because heterology can significantly improve the sensitivity of the immunoassay [41], the affinity of streptavidin and biotin (the affinity constant is 10^15^ L/mol) was higher than that of antigen and antibody (the affinity constant is 10^5−11^ L/mol) [24]. Under the optimized the H2-OVA and BAb concentration, the ionic strength and pH of the Bic-ELISA system, the LOD of the Bic-ELISA was 0.17 ng/mL, and the linear range was 0.25–25 ng/mL. Finally, in 15 authentic pollen samples on Taobao’s website, the acetamiprid concentrations were less than the detection range of the HPLC-MS/MS method (5–200 ng/g) and Bic-ELISA method (2–200 ng/g). To achieve the market level, we need more sensitive antibodies or detection techniques, which may require a lot of effort to screen antibodies, or more expensive detection instruments, or more sensitive analytical methods, such as electrochemical analysis, or fluorescence analysis. Although the validity of the method has not been verified by actual samples more directly, we are glad to see that the pollen samples we purchased are safe [The strictest standard of Acetamiprid MRL in agricultural products is cottonseed (MRL: 100 ng/mL), as established by Chinese regulation] for the environmental and human health.

## Figures and Tables

**Figure 1 molecules-24-01265-f001:**
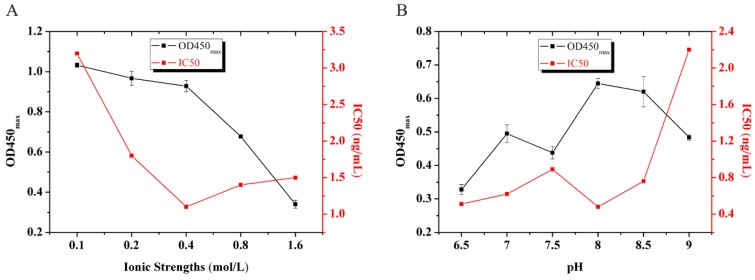
Effect of ionic strengths and pH values on the performance of the assay. The standard deviation (SD) of (**A**) were [0.1 (0.0111), 0.2 (0.0349), 0.4 (0.0282), 0.8 (0.0068), 1.6 (0.0202)]; The SD of (**B**) were [6.5 (0.0153), 7.0 (0.0267), 7.5 (0.0183), 8.0 (0.0151), 8.5 (0.0450), 9.0 (0.0086)].

**Figure 2 molecules-24-01265-f002:**
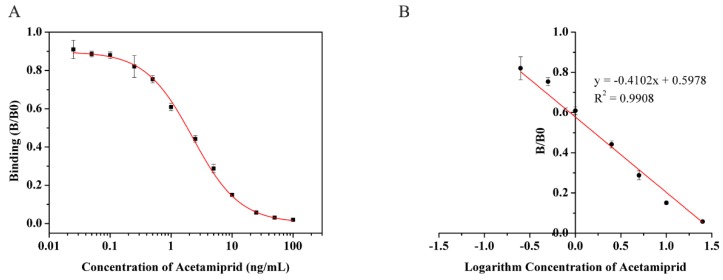
Detection curve of acetamiprid by biotinylated indirect competitive enzyme-linked immunosorbent assay (Bic-ELISA) (*n* = 5). a: binding curves of the detection, b: the detection line converted from (a). The SD of (**A**) were 0.0475, 0.0147, 0.0179, 0.0571, 0.0197, 0.0194, 0.0179, 0.0219, 0.0071, 0.0022, 0.0034, and 0.0041, respectively. The SD of (**B**) were 0.0571, 0.0197, 0.0194, 0.0179, 0.0219, 0.0071, and 0.0022, respectively.

**Figure 3 molecules-24-01265-f003:**
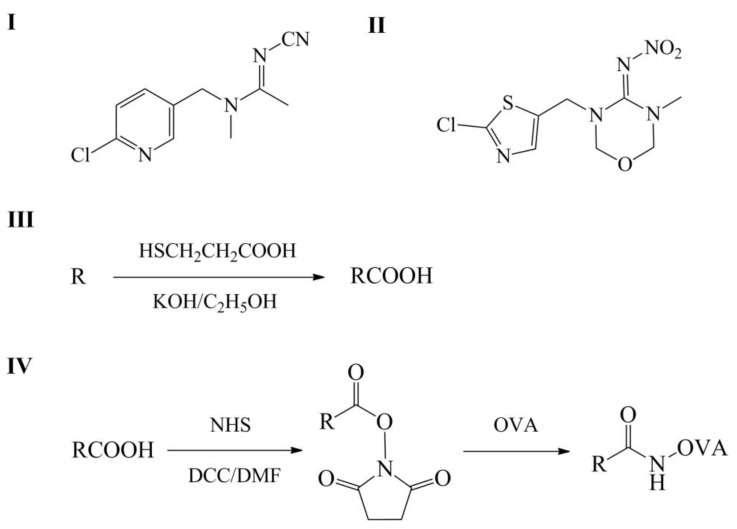
I: the chemical structure of acetamiprid; II: the chemical structure of thiamethoxam; III: the synthetic route to the acetamiprid hapten; IV: the active ester method.

**Table 1 molecules-24-01265-t001:** The cross-reactivity (CR) of acetamiprid toward other analogues.

Compound	Chemical Structure	IC50 (ng/mL)	CR (%)
Acetamiprid	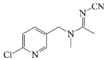	1.7	100
Thiacloprid	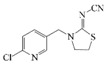	102.6	1.66
Thiamethoxam	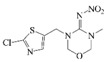	>400	<0.5
Imidacloprid	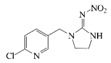	>400	<0.5
Dinotefuran	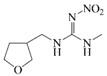	>400	<0.5
Nitenpyram	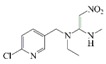	>400	<0.5
Clothianidin	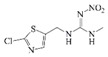	>400	<0.5
2,4-D	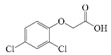	>400	<0.5

**Table 2 molecules-24-01265-t002:** Recovery studies of sample spiked with acetamiprid by Bic-ELISA.

Sample	Spiked (ng/g)	Intra-Assay (*n* = 3)			Inter-Assay (*n* = 3)		
		Mean ± SD ^a^ (ng/g)	Average Recovery (%)	RSD ^b^ (%)	Mean ± SD (ng/g)	Average Recovery (%)	RSD (%)
Camellia pollen	10	10.8 ± 1.0	108.0	9.3	11.1 ± 1.3	110.9	11.7
	20	18.7 ± 1.4	93.5	7.7	18.1 ± 1.3	90.5	7.2
	50	46.0 ± 2.7	92.0	5.9	44.7 ± 3.3	89.4	7.4
Lotus pollen	10	10.8 ± 1.0	107.6	9.0	10.6 ± 1.1	105.8	10.4
	20	16.2 ± 1.8	81.1	10.9	17.3 ± 1.2	86.7	6.9
	50	44.0 ± 2.1	88.0	4.8	42.7 ± 2.7	85.4	6.3
Rape pollen	10	10.4 ± 1.1	104.2	10.6	10.5 ± 0.9	104.9	8.6
	20	21.1 ± 1.8	105.5	8.3	20.3 ± 1.6	101.4	7.9
	50	52.6 ± 2.7	105.2	5.2	50.7 ± 3.1	101.3	6.1

^a^ SD: standard deviation. ^b^ RSD: relative standard deviation (*n* = 3).

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
