# Peer review of "Quantitative Determination of Acetamiprid in Pollen Based on a Sensitive Enzyme-Linked Immunosorbent Assay"

_molecules, 2019, doi:10.3390/molecules24071265_

Reviewer 1 Report

This work by  Fang et al presented a new method for high sensitivity detection of acetamiprid in pollen samples, filling the blank of a lack of efficient method in this field. The paper is organized in a methodological fashion with a straightforward structure and clear presentation of data. The results are convincing and well characterized. Though some minor improvements need to be made before it is ready for publication.

The author should provide a more detailed context for the use of IC50 and OD450 as their main criteria for evaluating the efficiency of this method.        

In fig 2, IC50 data points are lacking error bars. And please provide more info about the errors in fig2 and 3. Are they SD?          

line 214 to 215 seems to be a mix up.

For the last section, it is quite unsatisfying to see the lack of detectable data points for the commercial samples, making it hard to evaluate the effectiveness of this method in real world scenarios. The authors should consider acquiring pollen samples from known areas with relevant insecticide contamination to see if similar high sensitivity can be achieved with these natural samples.

In general, this work is of good quality and should be   considered favourably given the above comments adequately addressed.               

Author Response

Dear Reviewer:

We would like to thank you for your and editor’ critical comments and thoughtful suggestions to improve our manuscript. We have made careful modifications on the original manuscript. All changes made to the text are in red color. We hope the new manuscript will meet Molecules’s standard. Below you will find our point-by-point responses to your comments:

Comment: The author should provide a more detailed context for the use of IC50 and OD450 as their main criteria for evaluating the efficiency of this method. 

Response: Thanks for your suggestion. We have added a more detailed context for the use of OD450 in Line 88-89 and IC50 in Line 188-189 in our revised manuscript.

Comment: In fig 2, IC50 data points are lacking error bars. And please provide more info about the errors in fig2 and 3. Are they SD?    

Response: Thanks for your comments. The IC50 value is calculated by the one-dimensional regression equation. The X-axis is the logarithm of acetamiprid concentration, and the Y-axis is B/B0 (B and B0 are the absorbances of the analyte presence and absence, respectively). When the inhibition concentration reaches 50%, the concentration of acetamiprid is IC50 value, which was calculated, so there is no error bars.

We have added the values of SD in fig2 and 3 in Line 199-202 and Line 216-218 as follows: The standard deviation (SD)  of ‘‘Figure 2A’’ were [0.1 (0.0111), 0.2 (0.0349), 0.4 (0.0282), 0.8 (0.0068), 1.6 (0.0202)]; The SD of ‘‘Figure 2B’’ were [6.5 (0.0153), 7.0 (0.0267), 7.5 (0.0183), 8.0 (0.0151), 8.5 (0.0450), 9.0 (0.0086)].

The SD of ‘‘Figure 3A’’ were 0.0475, 0.0147, 0.0179, 0.0571, 0.0197, 0.0194, 0.0179, 0.0219, 0.0071, 0.0022, 0.0034, and 0.0041, respectively. And the SD of ‘‘Figure 3B’’ were 0.0571, 0.0197, 0.0194, 0.0179, 0.0219, 0.0071, and 0.0022, respectively.

Comment: line 214 to 215 seems to be a mix up.

Response: Thanks for your suggestion. We have deleted “Matrix Effects on Immunoassays” in our revised manuscript.

Comment: For the last section, it is quite unsatisfying to see the lack of detectable data points for the commercial samples, making it hard to evaluate the effectiveness of this method in real world scenarios. The authors should consider acquiring pollen samples from known areas with relevant insecticide contamination to see if similar high sensitivity can be achieved with these natural samples.

Response: Thank you very much for your suggestion. Because we have not considered the possible situation of the samples thoroughly enough, we have only prepared the pollen samples for sale on Taobao's website, but have not collected the pollen samples such as from areas known to be contaminated by pesticides, which makes it can not to determine the validity of the method more directly. Therefore, in the follow-up study, we will strive to obtain more abundant samples to verify the validity of analytical methods.

But at the same time, the results show that the residual of acetamiprid is below the maximum residue limit in agricultural products, as established by Chinese regulation, so we are pleased to see that pollen samples purchased from Taobao's website are safe.

We have added some discussion in the manuscript as follows: “Although the validity of the method has not been verified by actual samples more directly, we are glad to see that the pollen samples we purchased are safe [The strictest standard of Acetamiprid MRL in agricultural products is cottonseed (MRL: 100 ng/mL), as established by Chinese regulation] for the environmental and human health.” in Line 267-270.  

Reviewer 2 Report

The authors presented an important study, titled "Quantitative Determination of Acetamiprid in Pollen Based on a Sensitive Enzyme-Linked Immunosorbent Assay", about the detection of pesticides with novel technology, which developed a Bic-ELISA approach to quantify the levels of acetamiprid pesticides in pollen. The detection limit and the range for the Bic-ELISA were optimized in 0.17 ng/ml and 0.25-50 ng/ml, respectively. The results indicated that the measurements by the Bic-ELISA significantly harmonized with current analytical techniques. Therefore, the study concluded that the novel technology can be very helpful for the analysis of pesticides in pollen. This study is very important and highly related to the topic of the journal, of which the results can definitely help environmental researchers and chemists detect organic contaminants in plants. It is suggested that this manuscript should be considered for publication with some revisions. 

Keywords

Pesticide can be added as one of the keywords

Abstract 

Kindly add more significant results. For example, the reproducibility of the Bic-ELISA.

Introduction

Line 36-38, kindly add some references to show the plant pathogens caused by pest. References: Card et al. (https://link.springer.com/article/10.1071/AP07050 and https://doi.org/10.1111/1365-2664.12605) 

Line 39-44, to address the significance of the study, kindly discuss more about the human health impact of neonicotinoids. For example, human exposures to 2,4-D can via soil ingestion, inhalation, and dermal contact, which can result in some adverse health effects. References: Jennings et al. (https://doi.org/10.1016/j.jenvman.2014.07.020 and https://doi.org/10.1016/j.envint.2018.10.047)

Results and discussion

More discussion is required for 3.1. Verification of hapten 

To achieve the market level, the cost and portability are two significant factors, which need to be discussed. 

What's the environmental and human health implementation regarding the results?

Author Response

Dear Reviewer:

We would like to thank you for your and editor’ critical comments and thoughtful suggestions to improve our manuscript. We have made careful modifications on the original manuscript. All changes made to the text are in red color. We hope the new manuscript will meet Molecules’s standard. Below you will find our point-by-point responses to your comments:

Comment: Keywords: Pesticide can be added as one of the keywords

Response: Thanks for your suggestion. We have added “Pesticide” as one of the keywords.

Comment: Abstract: Kindly add more significant results. For example, the reproducibility of the Bic-ELISA.

Response: Thank you very much for your comments. Describe of the reproducibility of the Bic-ELISA was added in Line 23-24 as follows:

Analyte recoveries for extracts of spiked pollen (camellia pollen, lotus pollen, rape pollen) ranged from 81.1% to 108.0%, with intra-day relative standard deviations (RSDs) of 4.8% to 10.9%, and the average reproducibility were 85.4%-110.9% with inter-assay and inter-assay RSDs of 6.1% to 11.7%.

Comment: Introduction: Line 36-38, kindly add some references to show the plant pathogens caused by pest. References: Card et al. (https://link.springer.com/article/10.1071/AP07050 and https://doi.org/10.1111/1365-2664.12605

Response: Thanks for your suggestion. We have added two references to show the plant pathogens caused by pest in the Line 39.

Card, S.D.; Pearson, M.N.; Clover, G.R.G. Plant pathogens transmitted by pollen. Australas. Plant Path. 2007, 36, 455-461.

Van Rijn, P.C.J.; Wackers, F.L.; Cadotte, M. Nectar accessibility determines fitness, flower choice and abundance of hoverflies that provide natural pest control. J. Appl. Ecol. 2016, 53, 925-933.

Comment: Line 39-44, to address the significance of the study, kindly discuss more about the human health impact of neonicotinoids. For example, human exposures to 2,4-D can via soil ingestion, inhalation, and dermal contact, which can result in some adverse health effects. References: Jennings et al. (https://doi.org/10.1016/j.jenvman.2014.07.020 and https://doi.org/10.1016/j.envint.2018.10.047)

Response: Thanks for your comments. The references were added to address the significance of the study in the Line 45-47 as follows: However, the neonicotinoids can have an impact on human health, for example, human exposures to 2,4-D can via soil ingestion, inhalation, and dermal contact, which can result in some adverse health effects [13-14].

Jennings, A.A.; Li, Z. Scope of the worldwide effort to regulate pesticide contamination in surface soils. J. Environ. Manage. 2014, 146, 420-443.

Li, Z. A health-based regulatory chain framework to evaluate international pesticide groundwater regulations integrating soil and drinking water standards. Environ. Int. 2018, doi:10.1016/j.envint.2018.10.047.

Comment: Results and discussion: More discussion is required for 3.1. Verification of hapten 

Response: Thank you very much for your comments. We have added discussion for verification of hapten as follows: The results showed that acetamiprid hapten was successfully synthesized by one step method. The carboxylic acid moieties of hapten will facilitate the binding with carrier proteins to synthesize artificial antigens. (Line 179-181)

Comment: To achieve the market level, the cost and portability are two significant factors, which need to be discussed. 

Response: Thanks for your suggestion. Some discussion was added in the manuscript as follows: “To achieve the market level, we need more sensitive antibodies or detection techniques, which may require a lot of effort to screen antibodies, or more expensive detection instruments, or more sensitive analytical methods, such as electrochemical analysis, or fluorescence analysis.” in Line 264-266. 

Comment: What's the environmental and human health implementation regarding the results?

Response: Thanks for your comments. We have added some discussion in the manuscript as follows: “Although the validity of the method has not been verified by actual samples more directly, we are glad to see that the pollen samples we purchased are safe [The strictest standard of Acetamiprid MRL in agricultural products is cottonseed (MRL: 100 ng/mL), as established by Chinese regulation] for the environmental and human health.” in Line 267-270.